# Nikkomycin Z—Ready to Meet the Promise?

**DOI:** 10.3390/jof6040261

**Published:** 2020-10-30

**Authors:** David J. Larwood

**Affiliations:** 1Valley Fever Solutions, Tucson, AZ 85719, USA; davidl@valleyfeversolutions.com; Tel.: +1-650-454-4126; 2College of Pharmacy, University of Arizona, Tucson, AZ 85721, USA; 3California Institute of Medical Research, San Jose, CA 95128, USA

**Keywords:** antifungal, nikkomycin Z, NikZ, combination therapy, endemic dimorphic fungi, opportunistic fungi, *Coccidioides* spp., *Candida albicans*, *Aspergillus fumigatus*

## Abstract

Nikkomycin Z (NikZ) has fungicidal activity against some fungal species which currently requires patients to endure chronic therapy, sometimes for years. This review highlights reports of NikZ activity against fungal species for which current therapeutics are still inadequate, as a potential roadmap for continuing investigation. The possibility of faster and more complete clinical resolution by using NikZ has attracted scientific attention for decades. NikZ inhibits chitin structure formation, which is important for fungi, but not found in mammals. NikZ raised no safety concerns in a human Phase 1 trial or in extensive toxicology studies. NikZ showed strong clinical benefit in dogs with natural *Coccidioides* infection. NikZ has protected animals against fatal infections of *Candida albicans*. NikZ provides high protection in synergistic combination with several agent classes against *Candida* and *Aspergillus* species.

## 1. Introduction

Nikkomycin Z (“NikZ”, Figure 1) was first discovered as an antifungal agent in the 1970s [1]. The interest in NikZ increased when it was shown to be a significant inhibitor of chitin synthase [2]. The interest increased further when NikZ proved to be fungicidal against endemic, dimorphic fungi, showing significant clinical benefits in mammals against *Coccidioides*, *Histoplasma*, and *Blastomyces* spp. [3]. NikZ has shown protection against *Candida albicans* [4]. In combination, a chitin synthase inhibitor combined with various antifungal agents can be significantly synergistic against a range of medically important fungi. As the chitin synthase inhibitor that is the most advanced towards possible human clinical trials, NikZ could help in human therapies. NikZ potentiates a variety of agents when used in combination against important fungi, notably including *Candida albicans* [4] and *Aspergillus fumigatus* [5]. This review surveys early through recent reports of the state of development of NikZ.

## 2. Need for New Antifungal Agents

Only a few hundred fungal species cause severe infections in humans [6]. The limited number of current and developing systemic antifungal therapeutics provide imperfect therapies against these serious fungal infections. The frequent reviews of the status and range of antifungal therapies emphasize that there is room for better agents, particularly with the continuing incidence of drug resistance among many pathogens [6]. It is not uncommon for a given agent to act against some fungal genus, species, or strain, and not against what might seem to be similar strains.

Rauseo et al. note that about “1.7 billion individuals suffer from fungal infections worldwide” [7], and invasive or systemic infections “carry substantial human morbidity, mortality and economic burden” [8]. Also, “more than 90% of all reported fungal-related deaths result from infections caused by species of *Cryptococcus*, *Candida*, *Aspergillus*, *Histoplasma*, and *Pneumocystis*” [9,10]. LIFE reports 1.6 million deaths, and over 300 million affected by serious fungal diseases [11]. Infections caused by *Aspergillus* spp. and *Candida* spp. are the frequent focus of drug development. Recent reviews examine the available systemic antifungal drugs [7,12,13,14,15,16]. Thirteen new molecules include four in Phase 3 clinical trials [12]. An FDA workshop on 4 August 2020 discussed therapies for fungal diseases and a companion workshop on 5 August 2020 focused on therapies in development for coccidioidomycosis [17], including NikZ [18,19].

The endemic dimorphic fungi include *Coccidioides, Histoplasma,* and *Blastomyces* spp. Goughenour et al. [20] note that a “survey of hospital records in the United States tallied over 6000 hospitalizations were caused by the endemic fungi as follows: histoplasmosis (3360), blastomycosis and coccidioidomycosis (2640)” [21]. In addition, the incidence of fungal infections is high among the immunocompromised hosts by chemotherapy or disease (organ transplants, HIV infections). Almost half of the states in the USA have significant incidence of endemic fungal infections [22]. A recent California study concluded that coccidioidomycosis costs the state $700M [23]. Similar studies show Arizona hospitalizations cost $70M annually, with the overall state burden on the order of $700M [18].

## 3. NikZ: Sources and Manufacturing

Nikkomycin Z was first reported by Dähn et al. and Bayer colleagues as a secondary metabolite produced by *Streptomyces tendae* [1]. Its bioactivity was recognized as a potential antifungal agent against plant pathogens, the chemical properties were studied, and the structure was determined. A related patent was approved rather quickly, suggesting a significant advance [24]. Dähn reported NikZ MICs of 1 µg/mL against four strains of plant pathogens *Mucor hiemalis* and *Rhizopus circinans* and showed a significant inhibition of chitin synthesis by 0.5 µg/mL of NikZ.

Hector, working for a Bayer subsidiary, found NikZ interesting in his studies of chitin synthase inhibitors, expanding the foundation for the further development of NikZ.

The direct synthesis of NikZ has proved challenging; Nikkomycin Z has been produced by organisms other than *Streptomyces tendae*, with no prospects for commercial scale manufacturing.

Stenland et al. reported a *S. tendae* strain engineered to suppress the production of NikX and increase NikZ production, differentiating a manufacturing impurity component that had been challenging to separate [25]. Stenland reported an updated process for “79% [NikZ] purification yield and >98% relative purity of the final product…suitable for further scale up and cGMP production.” [25] (p. 268).

## 4. NikZ Activity against Pathogens

Polyoxins and the related nikkomycin compounds showed inhibition of chitin synthase [2]. These proved to be useful tools to elucidate cellular mechanisms in fungal cell wall formation, maintenance, and susceptibility. Hector et al. observed that some of these compounds suppressed infections by *Coccidioides posadasii*, particularly NikZ, which led to promising in vivo studies, including an INDA (Investigational New Drug Application) and human Phase I trials, now moving towards Phase II. Nikkomycin Z is fungicidal against endemic dimorphic fungi [3].

In one of the more extensive reports of NikZ activity against a wide range of fungal species, Li and Rinaldi [26] examined NikZ in vitro antifungal properties alone and in combination with fluconazole or itraconazole, testing against 110 isolates of some 24 species (checkerboard method). Selected results are summarized here in Table 1. NikZ in combination showed synergistic activity against several important fungi and additive activity against others. The authors note a “marked synergism” between NikZ and ITRAconazole against *A. fumigatus* and *A. flavus* [26] (p. 1401).

This review will focus first on the mechanism of action and origins of NikZ, then review NikZ impact on *Coccidioides* since this is well studied, and then move to other endemic dimorphic fungi, and finally, opportunistic *Candida* and *Aspergillus*.

### 4.1. NikZ Mechanism of Action: Cell Wall

In medically important fungi, cell wall processes have been studied very closely, such as the different cellular responses to maintain cell wall integrity. Fungal cell walls include chitin, beta glucans, and cellulose as structural components surrounding the bilayer [27,28,29,30]. The amounts of these structural components differ significantly between different species.

In a typical cross section, a chitin component is illustrated as a first shell just surrounding the lipid bilayer membrane, surrounded in turn by a second enveloping layer composed of various glucan components. Chitin synthase inhibitors such as NikZ are fatally disruptive to some fungi, particularly the spherule-endospore phase of *C. immitis* [2,3] and presumably similar phases in *H. capsulatum* and *Blastomyces* spp., endemic dimorphic fungi susceptible to NikZ.

Zhang et al. [31] detail the fungal mechanisms involved in chitin synthesis, including microvesicles, the chitosomes, for intracellular transport. Fungi contain various chitin synthases, which activate differently during the fungal life cycle, and this timing has an impact on drug design. Studies of chitin in yeast cell walls have identified three chitin synthases, CSI, CSII, and CSIII, considered the most specific and potent of chitin synthase inhibitors [32]. Cabib et al. studied chitin synthase III activity using NikZ, a specific inhibitor of CSIII in a *chs2* mutant to study morphological changes and viability in *Saccharomyces cerevisiae* [33]. A 1983 study of Polyoxin D (related to NikZ) as an inhibitor of chitin synthesis in *C. immitis* concluded that chitin is important to maintain the structural integrity of the spherule phase [2].

Echinocandin drugs block synthesis or incorporation of beta glucans into fungal cell walls. A common fungal response to cell wall damage is production of excess chitin, typically reducing susceptibility to echinocandin drugs. Combining an echinocandin with a chitin inhibitor such as NikZ increases potency against *C. albicans* and *A. fumigatus*. [27,28].

### 4.2. NikZ In Vivo—Early Studies in Coccidioides

Early observations that NikZ is fungicidal against dimorphic fungi prompted particular attention targeting *Coccidioides* spp. [3,34]. Other pathogens are discussed below.

Hector, working for a Bayer subsidiary, filed a patent in 1987 for nikkomycin X and Z to treat mammals infected with fungi with >10% by weight chitin in the cell wall in the parasitic phase [3]. In vitro, immature spherules exposed to 0.1 or 0.4 mM NikZ or NikZ (50, 200 ng/mL) stopped fungal growth within 8 h. The low dose of 20 ng/mL stopped endosporulation but “a small percentage” of cells appeared only weakened.

The in vivo studies concluded that NikX and NikZ “appear to be potentially fungicidal.”

Challenging mice with 500 *C. immitis* arthrospores intranasally and treating them orally with NikZ or NikX, 100 mg/kg TID (300 mpkd, mg/kg/day) for 9 days gave “negligible” fungal burden, “sterilized,” in the lung, liver, and spleen, as shown in Table 2A. Increasing the challenge inoculation by a factor of 10 and lowering the dose to 75 mg/kg BID (150 mpkd) or TID (225 mpkd) showed significant efficacy. One infected group was not started on NikX therapy until after the first death, giving a late-intervention cohort of 8 from which 4 survived. Table 2B. A 10K CFU challenge terminated at 4 days showed no fungi in the lungs of mice treated with NikZ at 150 mg/kg/day (75 BID). Table 2C.

In 1990, Hector et al. extended the evaluation of NikZ and NikX in murine models against coccidioidomycosis, histoplasmosis, and blastomycosis [34]. Table 3 details coccidioidomycosis results. Other diseases are discussed later. Oral NikZ at 20 and 50 mg/kg BID for 10 days protected 100% of the animals against pulmonary coccidioidomycosis. NikX was significantly less effective in this model. Oral NikZ at 100 mg/kg/day (divided BID) reduced lung CFU by log_10_ 5.98. Half of this dose QD reduced burden by log_10_ 2.58, 80% of the QD dose (40 mg/kg/day) reduced burden by an additional log_10_ 2.51 (5.09 less than untreated). It was found that “divided doses are more effective.”

A murine model of meningocerebral coccidioidomycosis showed that oral NikZ at 50 mg/kg BID (100 mpkd) for 21 days reached the CNS, with 60% survival to end of test at day 65 [34]. Untreated animals survived ≤ 9 days. Systemic blastomycosis and histoplasmosis are discussed below.

Shubitz et al. [35] reported the murine model results of NikZ dose levels and timing against *C. posadasii*, modifying Hector et al.’s paradigm of fatal dose challenge by using instead a sub-acute lethal dose (500 arthrospores) and delaying treatment until the fungal burden was increased 10–100 fold, 120 h after intra-nasal infection. Drug administration was subcutaneous (10, 20, 40, and 80 mg/kg BID), for seven days. By the seventh day of treatment, significant fungal burden was prevented (5 of 6 subjects had no detectable CFUs).

When evaluating therapy duration and dose frequency, treating BID was found to be more effective than treating QD (21 days QD at 80 mpkd prevented fungal burden almost to the levels achieved in 7 days BID, 33% of the QD duration). Evaluating dose level and frequency, fungal burdens were similar after treating 7 days QD at 80 mpkd or BID at 20 mpkd (4× higher dose required with QD). Fungal CFU burden did not increase after therapy, which was evidence of a fungicidal impact, and extending therapy beyond 7 days did not improve the outcome. Fungal burdens evaluated 2 or 21 days after therapy were statistically indistinguishable, tested after 7 or 21 days QD therapy. Therefore, it was evident that a relatively brief therapy at a high enough dose and frequency can sharply suppress disease, a durable suppression consistent with a fungicidal effect. NikZ therapy completely prevented infection of the spleen, in contrast with 67% infection in the untreated group (no numbers reported), showing NikZ prevented dissemination. In a connected PK study, an s.c. dose of 40 mg/kg gave an AUC of 18.6 µg h/mL.

The authors concluded this suggested a potential human dose of 250–500 mg BID, 7.7–15.4 mg/kg/day (mouse equivalent 63–125 mg/kg/day).

Recently, a 2020 meeting abstract reported that NikZ was significantly effective in a model of murine disseminated coccidioidomycosis after IV inoculation with a fatal dose of *C. immitis* arthroconidia [36].

Addressing the natural incidence of coccidioidomycosis, a small trial in dogs with natural disease showed significant benefit from NikZ therapy [37]. Canine therapy generally follows human guidelines, with generally similar outcomes. For humans, IDSA guidelines suggest fluconazole for “3 to 6 months or longer, depending on the clinical response” [38]. When treating dogs with oral NikZ BID for only 60 days, 78% improved (7 of the 9 dogs that completed the study), and 33% (3 of 9) showed “resolution or near resolution of symptoms” after oral dosing at 250 mg BID for 10 ± 5 kg and 500 mg BID for 22.5 ± 7.5 kg dogs, a rough human equivalent of 2000 mg/day. These dogs had naturally occurring, fairly well-established disease (1, 1.5, 3, 8, and mostly 12 weeks since diagnosis). The subjects presented with various comorbidities [37].

In summary, NikZ is fungicidal against *Coccidioides* spp., useful at smaller doses if given more frequently (BID superior to QD), reaching maximum impact in these murine models by 7 days, with persistent benefit unchanged if measured at 2 or 21 days post therapy, all supportive of the interpretation that the effect is fungicidal against *Coccidioides*.

### 4.3. NikZ and Endemic Mycoses

#### 4.3.1. *Histoplasma capsulatum* Variety *Capsulatum*

Histoplasmosis is one of the more serious fungal diseases, particularly worldwide, and particularly in sensitive populations [10].

“*H. capsulatum* variety *capsulatum* is endemic in certain areas of North, Central and South America, Africa, Asia, and mostly the Ohio and Mississippi river valleys in the USA; cases have also been reported from Europe” [39]. Some forms are life threatening, while some cause few or no symptoms. IDSA recommended therapy for many presentations is itraconazole, typically 200 mg TID for 3 days, then QD or BID for weeks, months or years depending on severity and patient specific details. Particularly for more severe presentations, an initial therapy with amphotericin B (deoxycholate or lipid forms) for 1–2 weeks is common, followed by a step down to ITRA as noted [39].

As a brief introduction to the following few paragraphs, an early in vivo NikZ study showed a useful reduction in fungal burden in lung and spleen after oral therapy [34]. In general, NikZ doses could be found that suppressed fungal growth. A combination on NikZ with fluconazole gave increased suppression and survival [40]. NikZ showed a significant variability in MIC against a variety of strains of *H. capsulatum* [41].

In 1990, Hector et al. showed NikZ significantly suppressed growth of *H. capsulatum* G217B [34]. After intravenous infection with 1 × 10^7^ yeast-phase cells, oral NikZ at 5 or 20 mg/kg BID provided 100% protection and even 1 mg/kg BID protected 40% to day 14, losing 1 more on day 23, see Table 4A. Organ loads after 8.5 × 10^5^ CFU IV and treating 5 days showed useful burden reduction, although not sterilization. See Table 4B.

In a 1998 combination study, Graybill et al. treated an *H. capsulatum* strain with oral NikZ (2.5-25 mg/kg BID) and oral fluconazole (FCZ) (5 mg/kg BID) alone and in combination [40]. In immune competent ICR mice, the authors found “a strong inoculum effect on the dose response curve of nikkomycin Z, with diminished efficacy at fungal inocula of more than 10^6^ CFU” [40] (p. 2371). See Table 5A. NikZ reduced fungal burden more than FCZ, and a combination improved CFU reduction, particularly in the liver, see Table 5B. Testing in BALB/c nude mice, a combination of NikZ (10 mpk/day) and FCZ (20 mpk/day) “showed markedly prolonged survival” compared to treating with single drugs. The test strain was very sensitive to NikZ with MIC of 0.5 µg/mL.

In a 2000 study [41] of NikZ, amphotericin B (AmB) and ITRA against *H. capsulatum* var. *capsulatum,* tested in vitro against 20 strains, NikZ showed a wide range of MICs (4 to >64 µg/mL, with MIC_90_ ≥ 64 µg/mL). AmB MICs were tightly clustered around a median of 0.50 µg/mL and MIC_90_ 0.5 µg/mL, and MICs of itraconazole were ≤0.019 mg/mL for every isolate. Selecting from the 20 isolates, a first with low and a second with high sensitivity to NikZ were studied in vivo.

Ten days of therapy after intratracheal inoculation with low sensitivity isolate 1 (NikZ MIC ≥ 65 mg/mL) showed 100% survival after: (a) oral NikZ 200 mg/kg/day divided BID; (b) oral ITRA 150 mg/kg/day divided BID; or (c) AmB 2 mg/kg/dose QOD (ea. 2 days). A lower dose of (d) oral NikZ 40 mg/kg/day gave 70% survival, and the (e) NikZ 10 mg/kg/day group reached 0% survival on day 14. Untreated animals reached 0% on day 12. Fungal burdens in liver and spleen were similar in Amb and ITRA groups, somewhat higher in the high dose NikZ group (200 mg/kg/day), and several log_10_ orders higher in the less protective medium dose NikZ group (40 mg/kg/day). See Table 6.

Equivalent therapy after intratracheal inoculation with isolate 2 (inoculation with 500K CFU, 5× higher for the weaker strain) gave 100% survival after all therapies. Information about fungal burdens is sparse, but the limited data shows reduction consistent with the high survival. See Table 6.

To summarize, in the reports to date, treating histoplasmosis, NikZ dose levels in the general range of conventional itraconazole dose levels provided good survival but somewhat higher residual organ fungal burden when treating with NikZ. Combining NikZ and FCZ lowered fungal burden in liver more effectively than any single drug reported.

#### 4.3.2. *Coccidioides immitis* and *C. posadasii*

Coccidioidomycosis is the subject of continuing periodic reviews [42,43]. The endemic area is primarily the American Southwest, with half of the known cases in the region of Phoenix, Arizona, with various other centers throughout the Americas. The annual incidence of coccidioidomycosis is about 150K infections: 50K likely to produce an illness warranting medical attention, 10–20K diagnosed and reported, 2–3K produce pulmonary sequelae, 600–1000 disseminated disease, and 160 deaths. Although the risk of disseminated infection is around 1% of symptomatic infections, this is as high as 75% among high risk patients [38].

IDSA guidelines [38] for treating *Coccidioides* infections are divided into 19 categories of disease presentation, with an additional 7 categories for high risk populations. Antifungal treatment is recommended for patients who have significantly debilitating illness or low immunity. Except for pregnant adults, treatment is an oral azole, such as fluconazole (≥400 or even 800 mg daily). The duration of therapy is 3–6 months or longer depending on clinical response. Posaconazole, voriconazole, and amphotericin B formulations could be considered. In disseminated infections, only about 50% respond to available agents.

NikZ treatment shows promise in various *Coccidioides* animal models, discussed above in Section 4.2.

#### 4.3.3. *Blastomyces* spp.

Blastomycosis occurs most often in the Midwestern, southeastern and south-central USA and Canadian provinces that border the Great Lakes or St. Lawrence Seaway [44]. However, the incidence is increasing. Although usually localized to the lungs, 25–40% develop extra-pulmonary infection. Dissemination is most frequent in immunosuppressed individuals. Moderate to severe pneumonia require therapy and patients may benefit from therapy, given the high dissemination rate. Recommended therapy is similar to that for histoplasmosis: primary therapy of ITRA, 200 mg QD or BID for 6–12 months after a loading dose of ITRA 200 mg TID, and in more severe cases begin therapy with a brief course of amphotericin. Serum levels of ITRA should be confirmed after ≥2 weeks of therapy.

Current therapies: Fluconazole may have some benefit for CNS blastomycosis infections. POSAconazole may be useful, but there are no reports. VORIconazole has been used successfully, particularly for refractory blastomycosis [44].

In NikZ development, Hector et al. evaluated NikZ against blastomycosis in mice and showed oral NikZ therapy with 20 or 50 mg/kg BID conferred 100% survival after mice were inoculated IV with a fatal dose *B. dermatitidis* [34]. In a delayed therapy test (infect cohorts, delay all therapy until first death in any group), NikZ at 20 mg/kg BID gave 75% survival, the best of all therapies tested, consistent with observations that NikZ is fungicidal against dimorphic fungi [34] (p. 591). Lung fungal burdens were suppressed almost four log_10_ orders at the lowest 10 mg/kg.day BID dose (2.43 v 6.19 untreated), better than AmB, 1 mg/kg (log_10_ 3.09 CFU). NikZ sterilized lungs at 100 mg/kg.day (BID, residual log_10_ 0.37 ± 0.37). See Table 7.

Clemons et al. tested high NikZ doses for short periods against *B. dermatitidis* and found high efficacy [45]. NikZ in vitro had MIC of 0.78 µg/mL and MFC (minimum fungicidal concentration) of 3.1 µg/mL against *B. dermatitidis* ATCC 26199. In a murine intranasal challenge with 30K viable yeasts, therapy for 10 days conferred 100% survival after oral NikZ dosing BID at 200, 400 or 1000 mg/kg/day. As active control, amphotericin B (6.25 mg/kg/day) conferred 100% survival (undetectable CFU in lung). Itraconazole at 200 mg/kg/day (100 BID, gavage) conferred 90% survival during therapy, losing 30% in days 14–20 post therapy, with the highest residual lung CFU burden in this study, consistent with the fungistatic protection of azoles.

Few of the animals treated with NikZ and none of the amphotericin B group showed any signs of infection. In contrast, all the ITRA group and untreated survivors showed infection.

Testing the effect of duration of NikZ therapy showed 100% survival in all dose groups after therapy for ≥ 5 days, and significant survival after only three days therapy.

The authors found that “unlike itraconazole, nikkomycin Z was biologically curative after 10 days of treatment, a rate equivalent to that of parenterally administered amphotericin B and is likely to be much better tolerated.” [45] (p. 2028). See Table 8.

#### 4.3.4. *Sporothrix* spp.

*S. schenckii* and other species of this genus are important global endemic fungi. Most cases are localized to the skin and subcutaneous tissues [46]. Spontaneous resolution is rare. Dissemination is rare, and difficult to treat. Therapy is typically with itraconazole, 200 mg orally QD for 3–6 months. Patients not responding may be given higher doses of ITRA or alternatives [46].

Sporotrichosis is presently reaching epidemic proportions from Rio de Janiero to the Uruguay border [47]. In a recent study, 17 clinical isolates were tested in vitro. For the single strain of *S. globosa*, the NikZ MIC was 6.3–12.5 µg/mL. A combination with itraconazole (ITZ) gave an FIC of 0.37, strong synergism. Against the 6 strains of *S. shenckii sensu stricto*, two showed NikZ MIC of 50–100, one MIC of 200, and 3 MICs ≥400. In combination with ITZ, FIC’s ranged from 0.09 to 0.37, all showing strong synergism. Against the 10 strains of *S. brasiliensis*, one strain showed an MIC of 100, 1 of 200–400, and the rest mostly >400 µg/mL. FICs were mixed, with 2 ≤ 0.37 (strong synergism), on FIC of 2 (indifferent), and the rest mostly about 0.50–0.56, showing weak synergism. Looking at minimum fungicidal concentration and fractional fungicidal concentration, the *S. globosa* strain showed NikZ MFC of 100–200 µg/mL, and with ITRA showed strong synergism. Two of the *S. schenckii* strains showed a combination with strong synergism. The remaining 14 strains tested showed high MFC, FFC of 2, and indifferent synergism. The “study showed the potential use of NikZ for sporotrichosis treatment. Further studies are needed to understand the future application of this drug” [47].

### 4.4. NikZ and Opportunistic Mycoses

Opportunistic fungi, particularly *C. albicans* and *A. fumigatus* kill >600K people worldwide annually, putting people with compromised immune systems at particular risk [6,48]. Combination therapies show promise, but given the complexities and wide range of combinations, a detailed discussion will be held for a subsequent review.

#### 4.4.1. *Candida* spp.

*Candida* antifungal resistance is becoming more prominent. *Candida albicans* is the third most common species isolated in bloodstream infections in the USA [49]. *C. albicans* and other *Candida* species can become resistant to azoles by overexpressing cellular efflux pumps, and develop cross-resistance to all azoles and echinocandins [50].

Li et al. [26] reported NikZ MICs of 4 and 2 µg/mL for *C. albicans* and *C. parapsilosis* respectively, as well as a synergistic FIC when adding either FCZ or ITRA (see Table 1 above).

Becker [51] demonstrated a delay in the mortality onset when treating murine candidiasis with NikZ. Additional studies have reinforced reports of poor or limited NikZ efficacy against *C. albicans*.

In 1998, Hector [4] reported 100% survival for rats following an IV infection of *C. albicans.* Treating the animals with a continuous IV infusion of NikZ for 96 h at 33 mg/kg/day suppressed kidney CFU burden from log_10_ mean 5.37 to 4.23. Treating at 100 mg/kg/day gave a burden of 3.48, or about 2 logs reduction, with a similar burden treating at 330 mg/kg/day.

Combination therapy increases NikZ potency in some models. Early studies of NikZ identified synergistic benefit with papulacandin (PB), an early echinocandin, against *C. albicans* [52]. *Candida* was perceived as a poor prospect for NikZ therapy as the 1% chitin content of the yeast is low. In contrast, beta-glucan is high, at 39% of the cell wall. Suppressing beta-glucan formation with increasing levels of PB induced increased chitin levels, suggesting perhaps an organism response to weakening of the beta glucan cell wall component. Adding NikZ or NikX to PB led to significant protoplast destruction. See also [53,54].

#### 4.4.2. *Aspergillus* spp.

*Aspergillus* remains a target of great interest for antifungal drugs. NikZ MICs are high for *Aspergillus* spp. A number of combinations show therapeutic promise. These are diverse enough to warrant a subsequent review.

Table 1 above shows synergy of NikZ with ITRA in vitro against three *Aspergillus* species [7]. Chiou combined NikZ with micafungan, synergistic in vitro when used in combination against *A. fumigatus* (2 isolates) [55].

Verweij [56] demonstrated that caspofungin in combination with NikZ inhibits (FICI 0.15) *A. fumigatus* growth in vitro. Polyoxin D did not have a significant effect in this test system, nor did three acylureas known to decrease chitin synthesis. Verweij [26] noted that caspofungin in *A. fumigatus* reduced beta glucan and increased chitin, noting that endogenous chitanases provide at least two modes to degrade chitin, used here to visualize fungal hyphae and how these change with caspofungin therapy.

In vivo, Luque et al. [5] found “significantly greater potency” when combining oral NikZ (200 mg/kg QD) with s.c. MICA (micafungin) (3 mg/kg BID) for 10 days after murine IV infection with *A. fumigatus*. The combination gave 100% survival, with a significant reduction in fungal load in brain and liver. Each dose alone was suboptimal, with >50% deaths during the treatment period. Clemons et al. 2006 found adding s.c. NikZ (100 mg/kg BID) to s.c. MICA (1 mg/kg BID) significantly prolonged survival in a mouse model of pulmonary aspergillosis [57].

## 5. NikZ Safety

NikZ development has focused on PO delivery, with favorable but limited reports of effective continuous IV and sustained release formulations.

After finding no or insignificant toxicity in several animal safety studies, human Phase 1 studies confirmed the safety of NikZ, reporting pharmacokinetics after human single oral doses from 250 to 2000 mg [58].

Preclinical animal studies showed no adverse effect level (NOAEL) or no detectable toxicity after a canine 300 mg/kg single oral dose or chronic dose studies (oral, 28 days), or murine 1000 mg/kg (rats, oral QD, 6 months), with no significant histology or pathology concerns [58]. By allometric scaling, 1000 mg in a rat correlates with about 250 mg/kg in humans, 8 times the maximum tested human dose of 31 mg/kg reported by Nix, (oral 2000 mg/day for a 65 kg human, 143 pounds).

Fairly common doses of oral fluconazole used to treat coccidioidomycosis in humans are 400 or 800 mg/day, by molar equivalents roughly equivalent to 650 or 1300 mg of NikZ. In 2014, Shubitz et al. [35] concluded that a human oral dose of 500 to 1000 mg/day (divided into BID doses) could be a promising therapeutic range against coccidioidomycosis.

## 6. Conclusions

Nikkomycin Z (“NikZ”) is fungicidal against dimorphic fungi, effective against natural *Coccidioides* infection in dogs, and shows no safety concerns through a human Phase 1 trial. NikZ shows promise for treating *C. albicans* and (in combination) *A. fumigatus*. The impact against histoplasmosis may warrant further development, given the worldwide impact. Convenient oral dosing plus an intravenous formulation provide flexible clinical alternatives.

Recent renewed interest in NikZ studies should expand our understanding of fungal physiology and potential clinical benefits.

## Figures and Tables

**Figure 1 jof-06-00261-f001:**
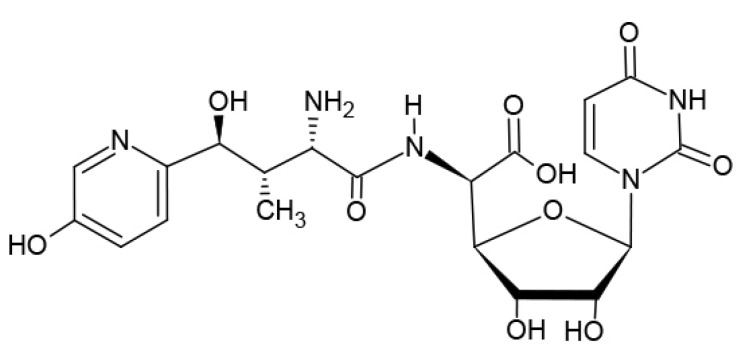
Nikkomycin Z (HCl). C20H25N5O10-HCl.

**Table 1 jof-06-00261-t001:** NIKZ MIC, FIC against various fungi [26].

Li Rinaldi—Selected	NikZ µg/mL		FIC	
	Range	MIC50	FCZ	ITRA
*Candida albicans*	<0.5–32	4	0.48	0.2
*Candida parapsilosis*	1–4	2	0.32	0.46
*Cryptoccus neoformans*	<0.5–<64	>64	0.72	0.49
*Aspergillus fumigatus*	>64	>64	>1	0.35
*Aspergillus flavus*	>64	>64	>1	0.35
*Aspergillus nidulans*	>64	>64	>1	0.52
*Coccidioides immitis*	1–16	4	0.44	0.47
*Bipolaris* spp.	>64	>64	0.64	NA
*Exophilalia dermatitiditis*	>64	>64	>1	0.56
*Ochroconi humicola*	4	4	>1	>1
	Synergistic < 0.5, Additive 0.5–1			

**Table 2 jof-06-00261-t002:** Early studies of NikZ (and NikX) against *Coccidioides immitis* [3]. mpk= mg/kg (**2A**): Initial challenge, 100 mg/kg NikX, NikZ TID, “Neg” = negative or negligible, (**2B**): mpk/day, 10x challenge, treat with 75 mg/kg BID or TID, treat 9 days, “GR” greatly reduced, “PD” prevented dissemination; (**2C**): mpk/day, 20× initial 500 CFU challenge, brief therapy, NikZ 75 mg/kg BID.

**(2A)**					
	**Oral TID 100**				
	mpk/day	Lung	Liver	Spleen	
Control	0	5.766	2.256	1.594	
NikX	300	Neg	Neg	Neg	
NikZ	300	Neg	Neg	Neg	
500 CFU Nares, +1 day, treat 9 days (oral, TID), +9 days					
**(2B)**					
	**BID 75**	**Survive**	**Lung**	**Liver**	**Spleen**
Control	0	20%			
NikX	150	90%	“GR”	“PD”	“PD”
NikZ	150	100%	“GR”	“PD”	“PD”
4800 CFU Nares, +2.2, treat 9 days (oral), +1 or 19					
Control (untreated): 80% mortality by day 11					
	TID 75	Survive			
NikX	225	50%			
After first untreated death, treat rest of group					
**(2C)**					
	**BID 75**				
Control	0	Lung: many in parasitic phase, early disease			
NikX	150	No intact fungal cells, a few abberant			
NikZ	150	No fungi, limited inflammation			
10K CFU Nares, +2.2, treat 4 days (oral), +1					

**Table 3 jof-06-00261-t003:** Pulmonary coccidioidomycosis (**3A**) survival, (**3B**) organ burden [34]

**(3A)**				
	**BID, Oral**		**Last**	
	mpkdy	survival	death (d)	
Control	0	0%	13	
NikX	10	12%	10	
NikX	40	0%	12	
NikX	100	45%	9	
NikZ	10	33%	9	
NikZ	40	100%	none	
NikZ	100	100%	none	
CFU 5–10K, treat day 2–11, +17, *n* = 10 mice				
**(3B)**				
	**Oral**		**Lung Log_10_**	**From**
	mpk/day		Mean CFU	control
Control	0		6.35	0
FCZ	5	BID	3.77	2.58
NikZ	100	BID	0.37	5.98
Control	0		6.21	0
NikZ	40	BID	1.12	5.09
NikZ	50	QD	3.63	2.58
CFU 4K, treat day 2–6, wait 2, *n* = 8 mice				

**Table 4 jof-06-00261-t004:** IV disseminated histoplasmosis (**4A**) survival, (**4B**) organ burden [34].

**(4A)**					
	**BID, oral**		**Last**		
	mpk/day	survival	death (d)		
Control	0	0%	20		
NikZ	2	30%	23		
NikZ	10	100%	none		
NikZ	40	100%	none		
CFU 10M *H. capsulatum*, treat day 2–11, +29, *n* = 10					
**(4B)**					
	**BID Oral**	**Lung Log_10_**	**From**	**Spleen Log_10_**	**From**
	mpk/day	Mean CFU	control	Mean CFU	control
Control	0	6.2	0	5.54	0
FCZ	50	5.34	0.86	4.71	0.83
NikZ	40	5.25	0.95	5.16	0.38
NikZ	100	4.37	1.83	4.46	1.08
CFU 850K *H. capsulatum*, treat day 2–6, wait 2, *n* = 8 mice					

**Table 5 jof-06-00261-t005:** IV disseminated histoplasmosis (**5A**) inoculum finding, days mean survival, ICR mice, inoculated IV, after 2 days treat BID for 10 days, hold to 30 days, (**5B**) organ burden. ICR mice, inoculate, after 2 days treat BID for 7 days, hold 1 more day [40].

**(5A)**					
**mpk/day**	**None**	**5**	**6**	**10**	**6, 10**
CFU		NikZ	NikZ	FCZ	Nik+FCZ
6.40 × 10^7^	5.3	4.3		5.3	
1.70 × 10^7^	4.6	8.1		8	
1.70 × 10^6^	9		16	21	26
1.00 × 10^6^	8	21		24	
**(5B)**					
**Inoculum CFU**		**2.30 × 10^5^**		**7.00 × 10^5^**	
	mpk/day	Spleen	Liver	Spleen	Liver
None		1.98	1.28	10.89	60.26
NikZ	10	0.22	2.12		
NikZ	20	0.11	1.89	1.96	6.97
NikZ	40	0.12	2.64		
NikZ	100	0.07	0.71	0.44	10.16
FCZ	20	3.72	8.46	4.08	19.43
10F + 10N	10 + 10			0.5	2.79
10F + 50N	10 + 50			0.37	2.87

**Table 6 jof-06-00261-t006:** pulmonary (intratracheal) histoplasmosis [41]: survival, organ burden (log10 CFU), two isolates. Inoculate, +4, treat 10 days, +1 or +4, Inoculate: isolate 1–100K CFU (NikZ MIC ≥64 µg/mL), Isolate 2–500K CFU (NikZ MIC 4 µg/mL).

		Survival	Isolate 1—CFU		Survival	Isolate 2—CFU	
	mpk/Day	14 Day	Liver	Spleen	17 Day	Liver	Spleen
Control		0%	n/r	n/r	30%	5.9	
AmB	1	100%	4	2.2	100%		
ITRA	150	100%	4.3	2.4	100%	2.3	2.7
NikZ	200	100%	5.7	5.2	100%		
NikZ	40	70%	8.2	8.5	100%	3.1	2.6
NikZ	10	0%			100%		

**Table 7 jof-06-00261-t007:** A, B: IV disseminated blastomycosis (**7A**) survival, (**7B**) organ burden [34].

**(7A)**				
	**BID, Oral**		**Last**	
	mpk/day	survival	death (d)	
Control	0	0%	8	
NikZ	10	40%	25	
NikZ	40	100%	none	
NikZ	100	100%	none	
CFU 49K, treat day 2–11, +19, *n* = 10				
**(7B)**				
	**Oral**		**Lung Log10**	**From**
	mpk/day		Mean CFU	control
Control	0		6.19	0
FCZ	50	BID	5.63	0.56
AmB	1	QD	3.09	3.1
NikZ	10	BID	2.43	3.76
NikZ	40	BID	1.81	4.38
NikZ	100	BID	0.37	5.82
CFU 50K, treat day 2–6, wait 2, *n* = 8 mice				

**Table 8 jof-06-00261-t008:** pulmonary blastomycosis: organ burden (log_10_ CFU lung); day of first death (none in AmB or NikZ ≥ 5 day), Survival as of Day 14 (10 day groups, 1 day post therapy), Survival as of final day 35. HPßCD* is carrier vehicle for Itra administration [45].

	mpk/Day	log10 CFU (Survivors)			Day of 1st Death	Day 14	Day 35 Survival		
Therapy Days		3 Days	5 Days	10 Days			3 Days	5 Days	10 Days
None				4.95	10	58%			25%
HPßCD*				3.39	9	70%			40%
Itra	200			6.14	8	90%			60%
AmB (i.p.)	6.25			0		100%			100%
NikZ	200	5.4	3.8	0.36	33	100%	70%	100%	100%
NikZ	400	4.7	4.68	1.15	25	100%	60%	100%	100%
NikZ	1000	3.05	2.95	0.68	27	100%	90%	100%	100%
Intranasal, 30 K CFU +4 days					Day	Early	Survival Final		

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
