# Peer review of "Nikkomycin Z—Ready to Meet the Promise?"

_jof, 2020, doi:10.3390/jof6040261_

Round 1
Reviewer 1 Report
This is a well written review of an agent that has been around for quite some time despite never being shown to be active in humans. As such the paper does not really add much to the NikZ literature. The fact that there are no prospects of commercial synthesis development (lines 78 and 79) detracts from the interest in such a paper.
Specific comments
- Pg 2, line 59, NikZ targets the cell wall not the cell membrane.
- Please specify which form of the dimorphic fungi is most susceptible to NikZ- yeast/spherule or hypha.
- Although there is a section at the end of the paper that states the route of administration it should be specified in each of the studies discussed for each of the fungal pathogens.
- Pg 8, line 329 and refs 26 and 55-the author refered to in the text and cited refs is Verweij not Verwer.
Author Response
Thank you, you comments are well taken. From several reviews, the manuscript (attached) is significantly updated.
Review 23 Sep 2020
Received Larwood 5 Oct 2020
Response 18 Oct 2020
Thank you for the thoughtful and helpful comments.
We made changes you suggested, then from other comments undertook more extensive edits. We hope that your comments have been addressed adequately
We are aware that Nikkomycin has not been the subject of much recent literature. As for adding to the literature, we hoped a review would assist with perspectives. We expect some additional reports in the near future, manuscripts in preparation but not suitable for a review of published work.
As a pharmaceutical chemist, we concur that poor prospects for commercial synthesis is not helpful, noting however that even if something can be made by synthesis this does not mean it would be the cheapest or even best preparation. NikZ is made easily by fermentation, at significant and we expect large scale. That section of the manuscript has been revised a bit to hopefully make this more clear. (below, as revised)
For a bit more perspective, many drugs are made by microbial fermentation at very large scale (penicillin, griseofulvin, amphotericin B). I share a few more notes below if you would like more detail.
This is a well written review of an agent that has been around for quite some time despite never being shown to be active in humans. As such the paper does not really add much to the NikZ literature. The fact that there are no prospects of commercial synthesis development (lines 78 and 79) detracts from the interest in such a paper.
Specific comments
- Pg 2, line 59, NikZ targets the cell wall not the cell membrane.
- Please specify which form of the dimorphic fungi is most susceptible to NikZ- yeast/spherule or hypha.
- Although there is a section at the end of the paper that states the route of administration it should be specified in each of the studies discussed for each of the fungal pathogens.
- Pg 8, line 329 and refs 26 and 55-the author referred to in the text and cited refs is Verweij not Verwer.
Specific comments 1 and 4 – thank you, corrected.
- (original 97-98)
Chitin synthase inhibitors such as NikZ are fatally disruptive to some pathogens, particularly the spherule-endospore phase of C. immitis [2, 3] and presumably similar phases in H. capsulatum and B. dermatitidis, endemic dimorphic fungi sensitive to NikZ.
3.Thank you for the comment. I added the following in the conclusion. (lines 357-360)
Nikkomycin Z (”NikZ”) is fungicidal against dimorphic fungi, effective against natural Coccidioides infection in dogs, and shows no safety concerns through a human Phase 1 trial. NikZ shows promise for treating C. albicans and (in combination) A. fumigatus. The impact against histoplasmosis may warrant further development, given the worldwide impact. Convenient oral dosing plus an intravenous formulation provide flexible clinical alternatives.
More broadly, thank you for the reminder. I added route to hopefully all reports.
Line 140 – add “orally”. Similarly in line 148 “Oral” NikZ …, 152, 209, 215-216, 340, 352-360
Line 317 IV
Line 344 s.c.
“Synthesis” comment, more detail:
Changes: (was 72-79)
Stenland et al. reported a Streptomyces tendae strain engineered to suppress the production of NikX and increase NikZ production, differentiating a mixture that had been challenging to separate [24]. Nikkomycin Z also has been produced by other organisms, and direct synthesis of NikZ has proved challenging, neither source in quantities that suggest a commercial scale opportunity. Stenland reported an updated process for a good NikZ fermentation yield, “79% purification yield and >98% relative purity of the final product … suitable for further scale up and cGMP production.” [24, p. 268].
Background: Microbial fermentation:
Many commercial drugs are made by microbial fermentation and are not practical for synthesis. In antifungals, amphotericin B is a good example. The molecule is quite complex. There are reports of synthetic preparation but so far as we know these are not used at commercial scale. Fermentation is fairly simple. I know from researching producers for NikZ that the list of fermented commercial pharmaceutical products is sizeable, including penicillin and griseofulvin. Wikipedia is not a deep citation, but see “Industrial Fermentation” and “Secondary Metabolites”. A PubMed search for “microbial fermentation” lists 19,361 results. Adding “secondary metabolites” narrows the list to 419 results.
For a potential entry point, the following appears it may be useful.
Bertrand S, Bohni N, Schnee S, Schumpp O, Gindro K, Wolfender JL. Metabolite induction via microorganism co-culture: a potential way to enhance chemical diversity for drug discovery. Biotechnol Adv. 2014 Nov 1;32(6):1180-204. doi: 10.1016/j.biotechadv.2014.03.001. Epub 2014 Mar 17. PMID: 24651031.
Reviewer 2 Report
In this review, the author provides an overview of in vitro and in vivo activities of Nikkomycin, thereby focusing on Nikkomycin Z as a novel antifungal therapy. Overall, although of general interest, I find this review very chaotic – it lacks structure. I recommend starting with a proper introduction, focusing on nikkomycin structure, in relation to other polyoxins or other nikkomycin derivatives (nikZ, nikX), as well as on its mode of action and specificity toward chitin synthase classes. Next, I recommend splitting the part on Nikkomycin activities in two parts: in a first part, nikkomycin activity alone should be described, both in vitro (including an overview table; also containing data on drug-resistant pathogens) and in vivo. The preclinical in vivo overview table should contain following info: model, pathogen (species, drug resistance profile, applied fungal burden), NikZ treatment regimen, primary outcome, reference. In a second part, nikkomycin-based combinations should be described. As a final part, clinical phase data/safety data can be presented.
Specific comments:
Abstract, first sentence: it is not the fungal species but patients that are suffering from the infections who need therapy. Moreover, include clear focus of the review in the abstract.
Line 30-32: explain imperfectly arrayed? Meaning not systematically tested?
Line 90-92 – echinocandins and their mode of action should be introduced properly
Line 108-110 – omit – this is not relevant in view of the review’s topic
Table 1 should be divided in two tables: one focusing on nikkomycin alone, another table focusing on potential synergy between nikkomycin and other antifungal drugs, as explained above. When comparing Nik with the other antifungal drugs, provide information and references of these specific drugs.
Line 132 – define ‘highly chitinous fungi’
Line 173 – explain – what other projections? Provide references.
Author Response
You comments are well taken, thank you.
Responding to inputs from reviewers, I attach a revised manuscript, redlined from the original submission. I tried to address each of your comments.
Review 23 Sep 2020
Received Larwood 5 Oct 2020
Response 18 Oct 2020
Thank you for your helpful comments.
In this review, the author provides an overview of in vitro and in vivo activities of Nikkomycin, thereby focusing on Nikkomycin Z as a novel antifungal therapy. Overall, although of general interest, I find this review very chaotic – it lacks structure. I recommend starting with a proper introduction, focusing on nikkomycin structure, in relation to other polyoxins or other nikkomycin derivatives (nikZ, nikX), as well as on its mode of action and specificity toward chitin synthase classes. Next, I recommend splitting the part on Nikkomycin activities in two parts: in a first part, nikkomycin activity alone should be described, both in vitro (including an overview table; also containing data on drug-resistant pathogens) and in vivo. The preclinical in vivo overview table should contain following info: model, pathogen (species, drug resistance profile, applied fungal burden), NikZ treatment regimen, primary outcome, reference. In a second part, nikkomycin-based combinations should be described. As a final part, clinical phase data/safety data can be presented.
Outline as submitted
Introduction
Need for better antifungals (lines 32-64 = 423 words) pointing to 3 recent reviews to frame the perspective for readers who may be newer to the field
Sources and Manufacturing (line 66-84 = 207 words) is intended to remind readers briefly of the origins of NikZ, and note the most recent publication regarding promising manufacturing. This section is revised some after other reviewer comments.
NikZ activity against pathogens (line 85)
This starts with a mention of articles that delve into your suggestion of structure / activity review. I trimmed a longer discussion along the lines of your suggestion.
Your comments suggest structure improvement for the review. An early draft of this article looked much like what you suggest, in about twice the allowed space. I had also divided out combination therapy from mono-therapy, plus a discussion of efforts to find new molecules based on SAR and biological studies. In paring to fit the space, I had to start with less detail on each point and found that citing the same article in 3 or 4 sections introduced other challenges (and more words).
I settled on a structure following type of organism, particularly since sensitivity to NikZ is high for 3 of the 5 typical endemic fungal species. The high-impact opportunistic fungi, particularly Candida and Aspergillus, are for the most part not particularly sensitive to NikZ but are sensitive in various combinations.
Combination studies seemed better incorporated in the flow of NikZ testing against each pathogen.
Safety studies proved to be relatively brief, collected in section 5.
Your table structure is a good idea. A challenge in such a structure is that different authors present different information, infection challenge may have varying impact by route, strain – usually presented in a context that makes sense in a line of reports but may not be harmonized well against different research.
Troubled by the “chaotic” comment, and comments from other reviewers,
I reduced original Table 1 to a much shorter version copying selectively from Li and Rinaldi, with some commentary on key data. The Rauseo component making the bulk of Table 1 is gone. My intent was to highlight promising potential benefits of NikZ that have been only partially explored in other reports in this review, and have faded from the memory of many.
Line 30-32: “imperfectly arrayed” was intended as a bit of creative expression. Edited to “therapeutics provide imperfect therapies against”
Line 90-92: edited to typically reducing susceptibility to echinocandin drugs, which typically block synthesis or incorporation of beta glucans, another important cell wall component.
Lines 108-110:
108 morphological changes and viability in Saccharomyces cerevisiae [31]. A 1983 study of Polyoxin D
109 (related to NikZ) as an inhibitor of chitin synthesis in C. immitis concluded that chitin is important in
110 maintaining the structural integrity of the spherule phase [2].
This would seem to support the request for more information about mechanism, also as an early report by the group that first recognized NikZ as potentially interesting as a therapeutic in mammals.
Table 1: Discussed a bit above. The suggested two tables are what is found in the literature, combined here to highlight perspective and similarities that may be easy to overlook.
Line 132: “highly chitinous” is a shorthand from other sources. Revised here to:
nikkomycin X and Z to treat mammals infected with fungi with >10% by weight chitin in the cell wall [3]. (copied from the reference)
Line 173: “other projections” deleted.
Round 2
Reviewer 2 Report
The structure of the review has improved.